# RANDOM GRAPH ASYMPTOTICS FOR TREATMENT EFFECT ESTIMATION IN TWO-SIDED MARKETS

## ABSTRACT

In two-sided markets, the accurate estimation of treatment effects is crucial yet challenging due to the inherent interference between market participants, which violates the Stable Unit Treatment Value Assumption (SUTVA). This paper introduces a novel framework that leverages random graph asymptotics to model and estimate treatment effects under network interference in two-sided markets. By incorporating a random graph model, we handle two-sided randomization by modeling customer interference within the potential outcome function as a function of graph topology and equilibrium dynamics, while capturing listing interference through the random graph structure. Our new estimation process provides asymptotically normal estimators with robust theoretical properties, suitable for large-scale market scenarios. Our theoretical findings are supported by extensive numerical simulations, demonstrating the effectiveness and practical applicability of our approach in estimating direct and indirect causal effects within these complex market structures.

## 1 INTRODUCTION

Two-sided markets have been a fundamental focus in microeconomics for several decades, playing a crucial role in various economic and digital platforms. In a two-sided market, two distinct groups of participants interact, with each group's success dependent on the presence and activity of the other. In the context of two-sided markets, such as online platforms or rental services, the two primary groups are typically customers and listings (e.g., products, services, or properties). Customers arrive sequentially and make choices from the listings that are available at the time of their arrival. Once a customer books a listing, that listing becomes unavailable for a period while it is being replenished. This temporary unavailability creates dynamic competition among customers as they vie for listings, and it also introduces a time-sensitive element to decision-making in the market. This structure creates complex interdependencies between the two sides, which present significant challenges in accurately estimating causal effects within two-sided markets.

Estimating causal effects within two-sided markets is a critical challenge across various fields. In these markets, interventions affect interactions between participants in the market, and decisions on whether to launch the intervention across the entire market depend heavily on the accuracy of experimental estimates. However, these estimates are often compromised by biases and high variances due to interference between market participants. As participants interact and compete, the treatment assigned to one individual can influence the behavior of others, thereby violating the Stable Unit Treatment Value Assumption (SUTVA) (Imbens & Rubin, 2015). Previous studies have shown that interference in markets can lead to substantial biases (Johari et al., 2022) and potentially infinite variances as market size increases (Li & Wager, 2022).

### 1.1 RELATED WORKS

Estimating treatment effects in two-sided markets has been a challenging area of research due to the interference between participants, which violates the Stable Unit Treatment Value Assumption (SUTVA). Several studies have attempted to address these challenges by developing models that account for this interference. One common approach is the bipartite experimental framework, where treatments are assigned to one set of market participants (e.g., sellers) and outcomes are observed

on another set (e.g., buyers) (Pouget-Abadie et al., 2019)(Zigler & Papadogeorgou, 2018). This framework is often represented as a bipartite graph connecting the two groups, which enables the modeling of market dynamics and interference (Harshaw et al., 2022). These studies laid the groundwork for designing experiments in two-sided markets but faced limitations when not only one side of the market receives treatment.

Recent advancements have focused on two-sided randomization designs, where both sides of the market receive treatment and control (Masoero et al., 2024). The predominant approach involves using a continuous Markov chain model combined with a mean field limit analysis (Johari et al., 2022). Wager & Xu (2021) and Li et al. (2022) leverage domain-specific structures within two-sided marketplaces and network routing to assess the bias of difference-in-means estimators across different experimental designs. Their model captures the dynamics of two-sided markets by analyzing the limiting case, where both the number of listings and the customer arrival rates approach infinity. Although this framework provides a solid foundation for analysis and introduces novel estimators, these estimators are particularly sensitive to the relative volumes of supply and demand. Biases can emerge due to the interconnected nature of participants, potentially leading to network effects and interference (Blake & Coey, 2014)(Cortez-Rodriguez et al., 2023)(Fradkin, 2019). Moreover, these estimators lack desirable asymptotic properties, underscoring the need for new approaches that can more effectively account for interference in two-sided markets.

In response to the limitations of traditional randomization techniques, a growing body of work has explored network-based interference models using exposure graphs. For example, Doudchenko et al. (2020) puts forward a framework that considers network structures in which edges probabilistically form between participants in bipartite structures, connecting treatments and outcomes through random formations. Erdos-Rényi random graphs are utilized for experimentation purposes to simulate network structures with interference. Cai et al. (2023) proposes an independent set design, which partitions a network into two sets: the independent set (non-interfering units) and the auxiliary set (interfering units), helping to control bias and variance in causal estimators by separating treatment and interference effects. Zheleva & Arbour (2021) further discusses techniques to mitigate interference in network-based randomized controlled trials, including block designs, chain graphs, and abstract ground graphs, which model the relationships between entities in the networks.

Utilizing random graphs to estimate causal effects is particularly valuable in networks with interference, such as two-sided markets, where traditional randomized experiments often fail to accurately estimate causal effects due to spillovers between connected units. Our research focuses on a model in which units are represented as vertices on a graph, with edges connecting any two units if the treatment of one may influence the potential outcome of the other (Li & Wager, 2022). This method has gained traction for its ability to model network interference, particularly in settings where the interference structure is complex. Appropriate random graph assumptions can facilitate more manageable analyses of treatment effects under network interference, leading to methodological advancements.

## 1.2 OUR CONTRIBUTIONS

By addressing the limitations of existing approaches, our contributions offer a significant advancement in the understanding and practical application of treatment effect estimation in two-sided markets. We introduce a new framework for analyzing experiments conducted in two-sided markets, and we propose a novel estimation process designed to achieve estimators with robust asymptotic properties. Our key contributions are as follows:

- We incorporate the random graph model proposed in Li & Wager (2022) to the two-sided markets, where the graph structure reflects the inherent competition and interference among market participants. To deal with two-sided randomization, we model the interference between customers within the potential outcome function as a function of both graph topology and equilibrium dynamics, and capture the interference between listings through the random graph model.

- We use the steady-state mass of listings to determine the graphon value, which represents the probability of two listings being connected, accounting for the interference between them. This allows for a realistic and analytically tractable analysis of treatment effects in two-sided markets.

- We identify key limitations in existing methods for treatment effect estimation in two-sided markets and introduce a new estimation process, yielding asymptotically normal estimators.

This framework not only advances the theoretical understanding of treatment effect estimation in two-sided markets but also provides practical tools for designing and analyzing experiments in these environments. The rest of the paper is organized as follows: Section 2 and 3 detail the formulation of our problem setting and random graph model, Section 4 presents our novel estimation process of the direct effect and indirect effect, Section 5 discusses the numerical analysis, and Section 6 concludes with implications and future directions.

## 2 PROBLEM FORMULATION: TWO-SIDED MARKETS

We are particularly interested in marketplaces where a booked listing becomes unavailable for a certain period before it becomes available again. Our model assumes a fixed number of listings, with the goal of exploring scenarios as the number of listings approaches infinity. Each arriving customer forms a consideration set from the available listings, and selects one to book or decides not to book at all, based on a choice model. Once a listing is booked, it becomes unavailable until its occupancy period ends.

Our primary interest lies in the steady-state average rate of bookings. We specifically focus on market interventions that alter the parameters governing customer choice probabilities. To test such interventions, we employ a two-sided randomization design: both customers and listings are randomly assigned to treatment or control groups. Each listing's consideration probability and utility for each customer are determined by their respective treatment condition.

**Listings:** The system consists of a fixed number $N$ of listings. Each listing $l$ has a type $\theta_l \in \Theta$. When an available listing is booked by an arriving customer, it becomes occupied, and an occupied listing of type $\theta$ remains occupied for an exponential time with parameter $\tau(\theta) = \tau\nu(\theta)$. Listings are indexed $i = 1, ..., N$, where each listing is randomly assigned a binary treatment $W_i \in \{0, 1\}$, $W_i \sim \text{Bernoulli}(a_L)$ for some $0 \le a_L \le 1$, and then experiences a potential outcome $Y_i \in \mathbb{R}$, where $Y_i$ represents the steady state rate at which bookings of listing $i$ are made by customers when time $t \to \infty$. We use a superscript '$N$' to denote quantities in the model with $N$ listings. Here we consider the regime where $N \to \infty$, allowing us to apply the mean field limit results as discussed in Johari et al. (2022) and the graphon asymptotic results outlined in Li & Wager (2022).

**Customers:** Each customer $j$ has a type $\gamma_j \in \Gamma$. Customers of type $\gamma$ arrive sequentially according to a Poisson process of rate $\lambda_\gamma$. Customers choose at most one listing to book and can choose not to book at all. Each customer is randomly assigned a binary treatment with probability $a_C$. The total arriving rate is denoted as $\lambda = \sum_{\gamma \in \Gamma} \lambda_\gamma$. Further, we assume that for each $\gamma \in \Gamma$, $\lim_{N \to \infty} \lambda_\gamma^{(N)}/\lambda^{(N)} = \phi_\gamma > 0$. Note that $\sum_\gamma \phi_\gamma = 1$.

**Booking Mechanism:** When customers arrive at a market, they form a consideration set of possible listings to book. Assume that each available listing of type $\theta$ is included in the arriving customer's consideration set independently with probability $\alpha_\gamma(\theta) > 0$ for a customer of type $\gamma$. To put it formally, suppose that customer $j$ arrives at time $T_j$. For each listing $\ell$, let $C_{j\ell} = 0$ if the listing is unavailable at $T_j$. If listing $\ell$ is available, let $C_{j\ell} = 1$ with probability $\alpha_{\gamma_j}(\theta_\ell)$, and let $C_{j\ell} = 0$ with probability $1 - \alpha_{\gamma_j}(\theta_\ell)$. Then the consideration set of customer $j$ is $\{\ell : C_{j\ell} = 1\}$. After the consideration set is formed, a choice model is applied to determine whether a booking is made.

We assume that a type $\gamma$ customer has utility $v_\gamma(\theta) > 0$ for a type $\theta$ listing. Let $q_{j\ell}$ denote the probability that arriving customer $j$ of type $\gamma_j$ books listing $\ell$ of type $\theta_\ell$. Assume that customers make choices according to the multinomial logit choice model:

$$q_{j\ell} = \frac{C_{j\ell}v_{\gamma_j}(\theta_\ell)}{\epsilon_{\gamma_j} + \sum_{\ell'=1}^{(N)} C_{j\ell'}v_{\gamma_j}(\theta_{\ell'})}$$

where $\epsilon_\gamma$ is the value of the outside option, corresponding to the circumstance that the customer doesn't book at all.

**Two-Sided Randomization of Treatment:** Interventions change the choice probability of listings by customers either through the consideration probabilities $\alpha$ or perceived utility $v$. The intervention is only applied when a treatment customer considers a treatment listing. Namely,

$$v_{\gamma,0}(\theta,0) = v_{\gamma,1}(\theta,0) = v_{\gamma,0}(\theta,1) = v_\gamma(\theta); v_{\gamma,1}(\theta,1) = \widetilde{v}_\gamma(\theta);$$
$$\alpha_{\gamma,0}(\theta,0) = \alpha_{\gamma,1}(\theta,0) = \alpha_{\gamma,0}(\theta,1) = \alpha_\gamma(\theta); \alpha_{\gamma,1}(\theta,1) = \widetilde{\alpha}_\gamma(\theta).$$

Additionally, we assume that both the value of the outside option and the parameter for the exponential occupancy time are not affected by the treatment condition. Namely,

$$\epsilon_{\gamma,1} = \epsilon_{\gamma,0} = \epsilon_\gamma; \nu(\theta,1) = \nu(\theta,0) = \nu(\theta).$$

**Interference:** Since each customer considers both treatment and control listings when deciding whether to book according to a multinomial logit choice model, there are dynamic interference between available listings, violating the Stable Unit Treatment Value Assumption (SUTVA). This interference introduces bias in estimation. Specifically, while the Horvitz–Thompson estimator, a natural estimator, is unbiased in the absence of interference, it may become biased in the presence of such interference.

**Causal Effect:** We seek to estimate the direct, indirect and total causal effects of the treatment, where $Y_i$ represents the steady state booking rate of listing $i$:

$$\bar{\tau}_{\text{DIR}}(a_L) = \sum_i E_{a_L}\left[Y_i(w_i = 1; W_{-i}) - Y_i(w_i = 0; W_{-i}) \mid Y(\cdot)\right],$$

$$\bar{\tau}_{\text{IND}}(a_L) = \sum_i \sum_{j \neq i} E_{a_L}\left[Y_j(w_i = 1; W_{-i}) - Y_j(w_i = 0; W_{-i}) \mid Y(\cdot)\right],$$

$$\bar{\tau}_{\text{TOT}}(a_L) = \bar{\tau}_{\text{DIR}}(a_L) + \bar{\tau}_{\text{IND}}(a_L)$$

We focus on estimating $\bar{\tau}_{\text{DIR}}(\pi)$ and $\bar{\tau}_{\text{IND}}(\pi)$.

## 3 RANDOM GRAPH MODEL

Investigating treatment effect estimation under random graph asymptotics has proven effective (Li & Wager, 2022). However, existing methods are limited to single-sided randomization, where only one set of units receives treatment. To extend this framework to two-sided randomization, we account for customer-side interference in the potential outcome function by incorporating both graph topology and equilibrium dynamics. Meanwhile, listing-side interference is modeled using a random graph approach. Specifically, we model the listings as the vertices of a random graph, with the interference graph represented as a random draw from a graphon. Given a set of regularity assumptions detailed below, we can estimate causal effects within a two-sided market framework in our setting. For direct effect estimation, standard estimators from the literature are both unbiased and asymptotically Gaussian. For indirect effects, our use of the PC balancing estimator demonstrates significant power.

To put the random graph model we use formally, we posit a graph with edge set $\{E_{ij}\}_{i,j=1}^N$ and vertices at the $N$ listings. The $i$th potential outcome may only depend on the $j$th treatment assignment if there is an edge from $i$ to $j$, that is, $Y_i(w) = Y_i(w')$ if $w_i = w_i'$ and $w_j = w_j'$ for all $j \neq i$ with $E_{ij} = 1$. We consider the following assumptions and demonstrate their validity in our setting of the two-sided markets.

**Assumption 1 (Undirected relationships)** The interference graph is undirected, that is, $E_{ij} = E_{ji}$ for all $i \neq j$.

From the market setting and customer choice model described in Section 2, the interference between listings is undirected, thus Assumption 1 holds true.

**Assumption 2 (Random graph)** The interference graph is randomly generated as follows: Each listing $i$ has a type $\theta_i \in \Theta$, and there is a symmetric measurable function $G_N : \Theta^2 \to [0,1]$ called a graphon such that $E_{ij} \sim \text{Bernoulli}(G_N(\theta_i, \theta_j))$ independently for all $i \neq j$ and $E_{ii} = 0$ for all $i \in \{1, 2, 3 \ldots N\}$.

To address dynamic interference within two-sided markets, we focus on the steady-state scenario. Referring to Johari et al. (2022), two-sided markets in our setting can be modeled with a continuous

Markov chain, where the state at any time reflects the number of available listings of each type. A mean field analog is proposed by examining the limit as both the number of listings and the customer arrival rate approach infinity. By scaling with the number of listings, a continuum of listings is obtained, and the state is represented by the mass of available listings in this mean field model. This system proved to be globally asymptotically stable and provide a precise characterization of the asymptotic steady state as the solution to an optimization problem. The mean field limit serves as the fluid limit of the finite market model, effectively approximating large markets.

Formally, the total mass of listings of type $\theta$ in the system is $\rho(\theta) > 0$ (note that $\sum_\theta \rho(\theta) = 1$). We represent the state at time $t$ by $\mathbf{s_t} = (s_t(\theta), \theta \in \Theta)$; $s_t(\theta)$ represents the mass of listings of type $\theta$ available at time $t$. The state space is:

$$\mathbf{S} = \{\mathbf{s} : 0 \le s(\theta) \le \rho(\theta)\}.$$

In an appropriate sense, there exists a unique steady state $\mathbf{s}^*$ to which all trajectories converge as $t \to \infty$ regardless of the initial condition. $\mathbf{s}^*$ is the unique solution to the following optimization problem:

$$\text{minimize} \quad W(s) \triangleq \sum_\gamma \sum_c \left( \lambda_{\gamma,c} \log \left( \epsilon_{\gamma,c} + \sum_\theta \sum_l \alpha_{\gamma,c}(\theta,l) \nu_{\gamma,c}(\theta,l) s(\theta,l) \right) \right)$$
$$- \tau(\theta) \sum_\theta \sum_l \rho(\theta,l) \log s(\theta,l) + \tau(\theta) \sum_\theta \sum_l s(\theta,l)$$

where $\lambda_{\gamma,1} = a_C \lambda$, $\lambda_{\gamma,0} = (1 - a_C)\lambda$; $\rho(\theta,0) = (1 - a_L)\rho(\theta)$, $\rho(\theta,1) = a_L\rho(\theta)$. Additionally, $c \in \{0,1\}$ is the treatment condition of the customers, and $l \in \{0,1\}$ is the treatment condition of the listings. From the definitions in Section 2, since interference only exists between listings that are available, we can set

$$G_N(\theta_i, \theta_j) = \sum_\theta s^*(\theta), \quad \forall i,j \in \{1,2,3...N\}, \quad i \ne j \tag{1}$$

as $t \to \infty$. Clearly, $G_N$ is a symmetric function, which satisfies the condition mentioned above. At this point, the random graph model in our paper is fully specified.

**Assumption 3 (Anonymous interference)** The potential outcomes only depend on the fraction of treated neighbors: $Y_i(w_i; w_{-i}) = f_i(w_i; \sum_{j \ne i} E_{ij} w_j / \sum_{j \ne i} E_{ij})$, where $f_i \in \mathcal{F}$ is the potential outcome function of the $i$th subject.

The rationality of Assumption 3 can be found in the Appendix. Under Assumption 2-3, we can also derive the specific form of the potential outcome function as a function of treatment status and proportion of treated neighbors, which represents the steady-state booking rate:

$$f_k(w, x) = \frac{\lambda}{N} \sum_i \sum_\gamma \frac{\phi_{\gamma,i} \alpha_{\gamma,i}(\theta_k, w) v_{\gamma,i}(\theta_k, w)}{\epsilon_{\gamma,i} + G_N \sum_{\theta'} (\alpha_{\gamma,i}(\theta', 1) v_{\gamma,i}(\theta', 1) x + \alpha_{\gamma,i}(\theta', 0) v_{\gamma,i}(\theta', 0)(1 - x))}. \tag{2}$$

Here, $G_N$ and $x$ characterize the incorporation of graph topology, while, as shown in (1), $G_N$ also reflects the equilibrium dynamics.

**Assumption 4 (Smoothness)** The potential outcome functions $f(w, x)$ satisfy

$$|f(w,x)|, |f'(w,x)|, |f''(w,x)|, |f'''(w,x)| \le B \tag{3}$$

uniformly in $f \in \mathcal{F}$, $w \in \{0,1\}$ and $x \in [0,1]$, where all derivatives of $f$ are taken with respect to the second argument.

The rationality of Assumption 4 is in the Appendix. This assumption states that small changes in the fraction of treated neighbors lead to small changes in the potential outcomes.

**Assumption 5 (Graphon sequence)** The graphon sequence $G_N(\cdot,\cdot)$ described in Assumption 2 (Random graph) satisfies $G_N(U_i, U_j) = \min\{1, \rho_N G(U_i, U_j)\}$, where $G(\cdot,\cdot)$ is a symmetric, non-negative function on $[0,1]^2$ and $0 < \rho_N \leq 1$ satisfies one of the following two conditions: $\rho_N = 1$ (dense graph), or $\lim_{N\to\infty} \rho_N = 0$ and $\lim_{N\to\infty} N\rho_N = \infty$ (sparse graph). In the case of dense graphs, we simply denote $G_N = G$.

Figure 1 illustrates the model setup under Assumption 1-5 on a small graph with 4 listings.

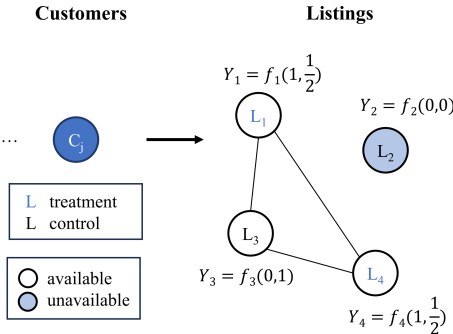

Figure 1: Random graph model setup

The following proposition offers a straightforward method for defining our target estimands within the specified random graph model. The direct effect measures how much $f$ changes with $w$, while the indirect effect is the derivative of $f$ with respect to $x$.

**Proposition 1** Consider a randomized trial under network interference satisfying Assumptions 1, 3, and 4, with treatment assigned independently as $W_i \sim \text{Bernoulli}(\pi)$ for some $0 < \pi < 1$. Let $N_i = \sum_{j\neq i} E_{ij}$ be the number of neighbors of subject $i$ in the interference graph. Conditional on the interference graph and the potential outcome functions, the estimands can be expressed as follows, where $B$ is the smoothness constant in Assumption 4 (Smoothness):

$$\bar{\tau}_{\text{DIR}} = \frac{\lambda}{N^2} \sum_i \sum_\gamma \sum_k \frac{\phi_{\gamma,i}(\alpha_{\gamma,i}(\theta_k,1)v_{\gamma,i}(\theta_k,1) - \alpha_\gamma(\theta_k)v_\gamma(\theta_k))}{\epsilon_{\gamma,i} + G_N \sum_{\theta'}(\alpha_{\gamma,i}(\theta',1)v_{\gamma,i}(\theta',1)\pi + \alpha_\gamma(\theta')v_\gamma(\theta')(1-\pi))}$$

$$+ \mathcal{O}\left(\frac{B}{\min_i N_i}\right),$$

$$\bar{\tau}_{\text{IND}} = \frac{1}{N} \sum_i \sum_\gamma \sum_k C\left(\pi\alpha_{\gamma,i}(\theta_k,1)v_{\gamma,i}(\theta_k,1) + (1-\pi)\alpha_\gamma(\theta_k)v_\gamma(\theta_k)\right)$$

$$+ \sum_i \sum_\gamma C\left(\sum_{\theta'}\alpha_{\gamma,i}(\theta',1)v_{\gamma,i}(\theta',1) - \alpha_\gamma(\theta')v_\gamma(\theta')\right) + \mathcal{O}\left(\frac{B}{\sqrt{\min_i N_i}}\right),$$

where $C = \frac{-\lambda\phi_{\gamma,i}G_N}{N\left(\epsilon_{\gamma,i}+G_N\sum_{\theta'}(\alpha_{\gamma,i}(\theta',1)v_{\gamma,i}(\theta',1)\pi+\alpha_\gamma(\theta')v_\gamma(\theta')(1-\pi))\right)^2}$.

## 4 ESTIMATION THEORY

### 4.1 DIRECT EFFECT

The direct effect captures the effect of a unit's treatment status on its own outcome. Under interference between available treatment listings and control listings as clarified in Section 2, the Horvitz-Thompson estimator (inverse propensity weighted, IPW, estimator) is unbiased for the direct effect conditionally on potential outcomes:

$$\hat{\tau}_{\text{DIR}}^{\text{HT}} = \frac{1}{n} \sum_i \frac{W_i Y_i}{\pi} - \frac{1}{n} \sum_i \frac{(1 - W_i) Y_i}{1 - \pi},$$

$$\mathbb{E}\left[\hat{\tau}_{\text{DIR}}^{\text{HT}} \big| Y(\cdot)\right] = \frac{1}{n} \sum_{i=1}^n \frac{\mathbb{E}\left[W_i Y_i(1, W_{-i}) | Y_i(\cdot)\right]}{\pi} - \frac{1}{n} \sum_{i=1}^n \frac{\mathbb{E}\left[(1 - W_i) Y_i(0, W_{-i}) | Y_i(\cdot)\right]}{1 - \pi}$$

$$= \frac{1}{n} \sum_{i=1}^n \frac{\mathbb{E}\left[W_i | Y_i(\cdot)\right] \mathbb{E}\left[Y_i(1, W_{-i}) | Y_i(\cdot)\right]}{\pi}$$

$$- \frac{1}{n} \sum_{i=1}^n \frac{\mathbb{E}\left[1 - W_i | Y_i(\cdot)\right] \mathbb{E}\left[Y_i(0, W_{-i}) | Y_i(\cdot)\right]}{1 - \pi}$$

$$= \bar{\tau}_{\text{DIR}}.$$

The Horvitz–Thompson estimator is consistent for the direct effect in both sparse and dense graphs, and it has a $1/\sqrt{N}$ rate of convergence regardless of the degree of the exposure graph (Li & Wager, 2022). Furthermore, we can establish a central limit theorem showing that natural estimator of the average treatment effect in the no-interference setting is asymptotically normal around the direct effect once interference effects appear.

**Theorem 2** Consider a randomized trial under network interference satisfying Assumptions 1–5. Suppose that the function $g_N(u) := \sum_\theta \min(1, G_N(u, \theta))$ is bounded away from 0, i.e.,

$$g_N(u) \geq c_l \quad \text{for any } u \in \Theta,$$

and that

$$\mathbb{E}[G(U_1, U_2)^k] \leq c_u^k \quad \text{for } k = 1, 2.$$

Finally, suppose that $\liminf \log \rho_N / \log N > -1$. Then the Horvitz–Thompson estimator has a limiting Gaussian distribution around the direct effect,

$$\sqrt{N} \left(\hat{\tau}_{\text{DIR}}^{\text{HT}} - \bar{\tau}_{\text{DIR}}\right) \xrightarrow{d} N\left(0, \pi(1 - \pi)\mathbb{E}\left[(R_k + Q_k)^2\right]\right),$$

where

$$R_k = \frac{\lambda}{N} \sum_i \sum_\gamma \frac{\phi_{\gamma,i}}{\epsilon_{\gamma,i} + G_N \sum_{\theta'} \left(\alpha_{\gamma,i}(\theta', 1) v_{\gamma,i}(\theta', 1)\pi + \alpha_\gamma(\theta') v_\gamma(\theta')(1 - \pi)\right)} \cdot$$

$$\left(\frac{\alpha_{\gamma,i}(\theta_k, 1) v_{\gamma,i}(\theta_k, 1)}{\pi} + \frac{\alpha_\gamma(\theta_k) v_\gamma(\theta_k)}{1 - \pi}\right),$$

$$Q_k = \mathbb{E}\left[\frac{G(U_k, U_j)\left(f_j'(1, \pi) - f_j'(0, \pi)\right)}{g(U_j)} \bigg| U_k\right].$$

### 4.2 INDIRECT EFFECT

We take advantage of the PC balancing estimator initially proposed in Li & Wager (2022) to estimate the indirect effect in a setting where the graphon $G$ admits a low-rank representation with rank $r$, that is, $G(U_i, U_j) = \sum_{k=1}^r \lambda_k \psi_k(U_i)\psi_k(U_j)$ for a small number $r$ of measurable functions $\psi_k : [0, 1] \to \mathbb{R}$. The low-rank condition quantifies an assumption that each unit can be characterized using a small number $(r)$ of factors. Qualitatively, since the only factor that characterize the listings is the type, the low-rank condition is naturally satisfied.

In practice, we don't have access to $\psi_k(U_i)$ directly. But if the graphon is low rank, then the adjacency matrix $E$ is a noisy observation of the low rank edge probability matrix, and its eigenvectors $\hat{\psi}_{ki}$ are approximately $\psi_k(U_i)$. PC balancing estimator can be calculated following the procedure:

1. Let $E$ be the adjacency matrix with 0 on the diagonal.

2. Eigen-decompose $E$: Let $\lambda_1, \lambda_2, \ldots, \lambda_r$ be the first $r$ eigenvalues of $E$ such that $|\lambda_1| \geq |\lambda_2| \geq \cdots \geq |\lambda_r|$. Let $\psi_k$ be the eigenvector corresponding to the eigenvalue $\lambda_k$.

3. Compute the PC balancing estimator

$$\hat{\tau}_{\text{IND}}^{\text{PC}} = \frac{1}{n}\sum_i Y_i\left(\frac{M_i}{\pi} - \frac{N_i - M_i}{1 - \pi} + \sum_{k=1}^r \hat{\beta}_k \psi_k(U_i)\right),$$

where $\hat{\beta}$ is determined by solving the following equations

$$\sum_i \psi_l(U_i)\left(\frac{M_i}{\pi} - \frac{N_i - M_i}{1 - \pi} + \sum_{k=1}^r \hat{\beta}_k \psi_k(U_i)\right) = 0,$$

for all $l = 1, 2, \ldots, r$.

The PC balancing estimator is both consistent and satisfies a central limit theorem, demonstrating its asymptotic normality (Li & Wager, 2022). Under certain regularity assumptions, the PC balancing estimator is asymptotically normal:

$$\frac{\hat{\tau}_{\text{IND}}^{\text{PC}} - \tau_{\text{IND}}}{\sqrt{\rho_n}\sigma_{\text{IND}}} \xrightarrow{d} \mathcal{N}(0, 1), \frac{\hat{\tau}_{\text{IND}}^{\text{PC}} - \bar{\tau}_{\text{IND}}}{\sqrt{\rho_n}\sigma_{\text{IND}}} \xrightarrow{d} \mathcal{N}(0, 1),$$

where $\sigma_{\text{IND}}^2 = \mathbb{E}[G(U_1, U_2)(\alpha_1^2 + \alpha_1\alpha_2)] + \mathbb{E}[g(U_1)\eta_1^2]/(\pi(1 - \pi)), \alpha_i = f_i(1, \pi) - f_i(0, \pi), b_i = \pi f_i(1, \pi) + (1 - \pi)f_i(0, \pi)$ and $\eta_i = b_i - \sum_{k=1}^r \mathbb{E}[b_i\psi_k(U_i)]\psi_k(U_i)$.

## 5 NUMERICAL STUDIES

In this section, we implement the estimation process developed in the previous sections to evaluate its performance under simulated scenarios. Our goal is to demonstrate the robustness and effectiveness of the proposed methodology for estimating causal effects in two-sided markets, considering both homogeneous and heterogeneous settings with network interference. The results provide insights into the practical applicability of our approach.

### 5.1 PARAMETERS

**Homogeneous Setting**: Consider a market with homogeneous customers and listings. We set $\epsilon = 1$, $\alpha = 0.5$, $a_C = a_L = 0.5$. Customers have utility $v = 0.315$ for control listings and $\tilde{v} = 0.394$ for treatment listings. $\lambda = \tau = 1$.

**Heterogeneous Setting**: Consider a market with heterogeneous customers. There is one listing type $\theta$ and two customer types $\gamma_1, \gamma_2$. We fix the size of the treatment utility increase such that $\tilde{v}_{\gamma_1}(\theta) = 1.25 \cdot v_{\gamma_1}(\theta)$ and $\tilde{v}_{\gamma_2}(\theta) = 1.25 \cdot v_{\gamma_2}(\theta)$. We set the ratio $v_{\gamma_2}(\theta)/v_{\gamma_1}(\theta) = 3$, with $v_{\gamma_1}(\theta) = 0.17, v_{\gamma_2}(\theta) = 0.51$.

### 5.2 DIRECT EFFECT

#### 5.2.1 SIMULATION WITH THE POTENTIAL OUTCOME FUNCTION

We simulate data as described in the homogeneous setting, for a graph with $N = 10000$ nodes generated via a constant graphon $G_N(u_1, u_2) = 0.867$, where any pair of nodes are connected with probability 0.867. The value of the graphon is calculated according to (1). We then generate treatment assignments as $W_i \stackrel{\text{i.i.d.}}{\sim} \text{Bernoulli}(\pi)$ with $\pi = a_L = 0.5$, and potential outcome functions as (2). Note that $\phi_{\gamma,0} = (1 - a_C)\phi_\gamma, \phi_{\gamma,1} = a_C\phi_\gamma$.

As proved in Section 4, the Horvitz-Thompson estimator is unbiased for the direct effect conditionally on potential outcomes, and is asymptotically normal around the direct effect. Hájek estimator is not exactly unbiased, but its ratio form makes it invariant to shifting all outcomes by a constant. Figure 2 shows the distribution of the estimators $\hat{\tau}_{\text{DIR}}^{\text{HT}}$ and $\hat{\tau}_{\text{DIR}}^{\text{HAJ}}$ across 2000 simulations. Here, the Hájek estimator has a better asymptotic variance than the Horvitz–Thompson estimator, which agrees with the case without interference.

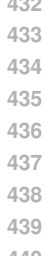
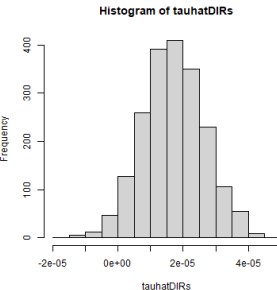
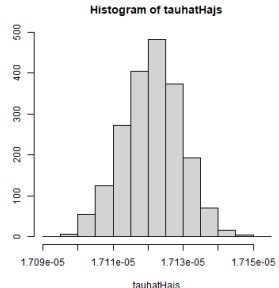

Figure 2: Histograms of the Horvitz–Thompson and Hájek estimator

### 5.2.2 SIMULATION WITH THE MARKET SETTING

We use another simulation approach of the direct effect to verify the accuracy of the result in Section 5.2.1. We simulate the process of customers arriving, choosing a listing, the listing being occupied, and the listing becoming available again.

According to simulation, in the homogeneous setting, $\hat{\tau}_{\text{DIR}}^{\text{HT}} = \hat{\tau}_{\text{DIR}}^{\text{HAJ}} = 1.122 \times 10^{-5}$. The standard error estimation for both the Horvitz-Thompson estimator and the Hájek estimator are $1.898 \times 10^{-6}$. The results of this simulation are of the same order of magnitude as in Section 5.2.1 and differ within the expected range, indicating that our simulation is reliable. The reason why two ways of simulation don't get exactly the same result is that simulation with the potential outcome functions depends on a asymptotic mean field model when $N \to \infty$, while simulation with the market setting uses a specific value of $N = 10000$. Similarly, in the heterogeneous setting, $\hat{\tau}_{\text{DIR}}^{\text{HT}} = \hat{\tau}_{\text{DIR}}^{\text{HAJ}} = 1.082 \times 10^{-5}$, and the standard error estimation for both the Horvitz-Thompson estimator and the Hájek estimator are $1.747 \times 10^{-6}$.

## 5.3 CONSERVATIVE INTERVALS FOR THE DIRECT EFFECT

We consider the conservative interval for the direct effect in the homogeneous setting. According to Li & Wager (2022), under Assumptions 1-5 and certain regularization assumptions, the Hájek estimator $\hat{\tau}$ satisfies a central limit theorem

$$\sqrt{N}(\hat{\tau} - \bar{\tau}_{\text{DIR}}) \implies \mathcal{N}(0, \sigma_0^2 + \pi(1-\pi)V),$$

where $V = \text{Var}[R_i] + 2\text{Cov}[R_i, Q_i] + \mathbb{E}[Q_i^2]$. Let $V_0 = \text{Var}[R_i]$. By Cauchy–Schwarz,

$$V \le V_0 + 2\sqrt{V_0\mathbb{E}[Q_i^2]} + \mathbb{E}[Q_i^2],$$

$$\frac{1}{2}\mathbb{E}[Q_i^2] \le \frac{\mathbb{E}[a_0^2(U_i)]}{\mathbb{E}[a_0(U_i)]^2}\mathbb{E}\left[f_i'(1,\pi) - f_i'(0,\pi)\right]^2$$

$$+ \sum_{k=1}^{K} \mathbb{P}[U_i \in I_k]\frac{\mathbb{E}[a_k^2(U_i)|U_i \in I_k]}{\mathbb{E}[a_k(U_i)|U_i \in I_k]^2}\mathbb{E}\left[f_i'(1,\pi) - f_i'(0,\pi)|U_i \in I_j\right]^2 \quad (4)$$

We use the standard error estimate from Section 5.2.2 for $V_0$, that is, $(\sigma_0^2 + \pi(1-\pi)V_0)/N = 3.599 \times 10^{-12}$. Since each listing has a type, the market naturally satisfies the setting of the disjoint communities model, where the interference effect is dominated by links between these disjoint communities, namely, different types of listings. Assume that $\mathbb{E}[f_i'(1,\pi) - f_i'(0,\pi)]^2 \le \tau_{DIR}^2$ and that all terms in (4) that depend on stochastic fluctuations can be controlled by considering these terms constant and then inflating the resulting bound by a factor 2. We get that

$$\mathbb{E}[Q_i^2] \le 8\tau_{DIR}^2.$$

The following chi-squared test will reject with probability at most $\alpha$ under the null hypothesis $H_0 : \tau_{\text{DIR}} = \tau_0$:

$$(\hat{\tau} - \tau_0)^2 \ge \frac{\Phi(1 - \alpha/2)^2}{N}(\sigma_0^2 + \pi(1-\pi)(V_0 + 2\sqrt{8V_0\tau_0^2} + 8\tau_0^2))$$

In our specific case, $\pi = 1/2$ and $N = 10000$, and we assume that $(\sigma_0^2 + \pi(1-\pi)V_0)/N = 3.599 \times 10^{-12}$. We can maximize the noise term by setting $\sigma_0^2 = 0$ and $V_0 = 4 \times 10000 \times 3.599 \times 10^{-12}$, so that the hypothesis test is fully specified. We obtain the following 95% confidence interval: $\tau_{\text{DIR}} \in (7.299165 \times 10^{-6}, 1.5365257 \times 10^{-5})$. In contrast, the unadjusted Gaussian confidence interval was $\tau_{\text{DIR}} \in (7.501121 \times 10^{-6}, 1.4939679 \times 10^{-5})$. Although interference do inflate the confidence interval of the direct effect, we are still able to reject the null that $\tau_{\text{DIR}} = 0$ at the 95% level. Figure 3 shows the intervals obtained for different significance levels $\alpha$.

### 5.4 INDIRECT EFFECT

With the PC balancing estimator proposed in Section 4.2, we get $\hat{\tau}_{\text{IND}}^{\text{PC}} = 9.68 \times 10^{-6}$ in the homoegeneous setting, and $\hat{\tau}_{\text{IND}}^{\text{PC}} = 1.375 \times 10^{-5}$ in the heterogeneous setting. Figure 4 shows the distribution of $\hat{\tau}_{\text{IND}}^{\text{PC}}$ across 100 simulations, which is approximately normally distributed.

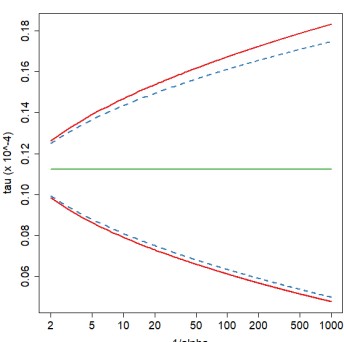

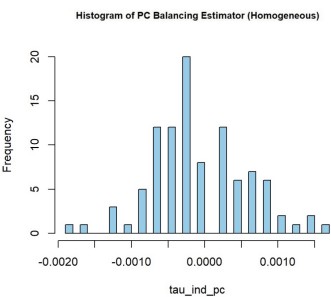

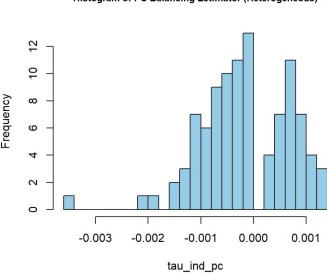

Figure 3: Level-$\alpha$ confidence intervals for the direct effect $\tau_{\text{DIR}}$. The dashed blue lines denote upper and lower endpoints of a basic Gaussian confidence interval, while the solid red curves denote endpoints of a confidence interval derived with inteference considered. The solid green line denotes the point estimate.

Figure 4: Histogram of PC Balancing Estimator across 100 simulations

## 6 CONCLUSION

This study advances the understanding of treatment effect estimation in two-sided markets by addressing the limitations of existing methodologies, particularly those affected by interference. By introducing a random graph model and novel estimation techniques, we provide a robust framework capable of handling the complexities of two-sided markets. Our simulations confirm the reliability of the proposed estimators, showing their potential for practical application in real-world scenarios. Future work may explore extending it to observational study settings, or adapting the framework for A/B testing scenarios. These contributions open new avenues for designing and analyzing experiments in two-sided markets, with implications for both theoretical research and applied economic strategies.

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

## A  APPENDIX

### A.1  DERIVATION OF THE POTENTIAL OUTCOME FUNCTION

By using the equilibrium described in Section 2 and by modifying the mean field limit result in Johari et al. (2022), the steady state rate at which bookings of listing $k$ of treatment condition $W_k \in \{0, 1\}$ are made by customers can be written as:

$$f_k(w, x) = \frac{\lambda}{N s^*(\theta_k, W_k)} \sum_{i \in \{0,1\}} \sum_{\gamma} \phi_{\gamma,i} p_{\gamma,i}(\theta_k, W_k | \mathbf{s}^*(a_C, a_L))$$

$$= \frac{\lambda}{N} \sum_{i} \sum_{\gamma} \frac{\phi_{\gamma,i} \alpha_{\gamma,i}(\theta_k, w) v_{\gamma,i}(\theta_k, w)}{\epsilon_{\gamma,i} + \sum_{j} \sum_{\theta'} \alpha_{\gamma,i}(\theta', j) v_{\gamma,i}(\theta', j) s^*(\theta', j)}$$

where $f_k(w, x)$ is the potential outcome function for listing $k$.

## A.2 RATIONALITY OF THE ASSUMPTIONS

**Rationality of Assumption 3:**

In the expression of $f_k(w, x)$, the only term affected by the treatment condition of the neighbors of listing $k$ is $s^*(\theta', j)$. According to Assumption 2 (Random graph), $\sum_{m \neq k} E_{km} w_m / \sum_{m \neq k} E_{km} = \sum_{\theta'} s^*(\theta', 1) / \sum_{j} \sum_{\theta'} s^*(\theta', j)$, and $\sum_{j} \sum_{\theta} s^*(\theta) = G_N$. When $\alpha$ and $v$ are both independent of the type of the listing,

$$\sum_{j} \sum_{\theta'} \alpha_{\gamma,i}(\theta', j) v_{\gamma,i}(\theta', j) s^*(\theta', j) = \sum_{\theta'} \alpha_{\gamma,i}(\theta', 1) v_{\gamma,i}(\theta', 1) s^*(\theta', 1)$$

$$+ \sum_{\theta'} \alpha_{\gamma,i}(\theta', 0) v_{\gamma,i}(\theta', 0) s^*(\theta', 0)$$

$$= \alpha_{\gamma,i}(\theta', 1) v_{\gamma,i}(\theta', 1) \sum_{\theta'} s^*(\theta', 1)$$

$$+ \alpha_{\gamma,i}(\theta', 0) v_{\gamma,i}(\theta', 0) \sum_{\theta'} s^*(\theta', 0)$$

$$= \alpha_{\gamma,i}(\theta', 1) v_{\gamma,i}(\theta', 1) G_N \frac{\sum_{m \neq k} E_{km} w_m}{\sum_{m \neq k} E_{km}}$$

$$+ \alpha_{\gamma,i}(\theta', 0) v_{\gamma,i}(\theta', 0) G_N \left( 1 - \frac{\sum_{m \neq k} E_{km} w_m}{\sum_{m \neq k} E_{km}} \right)$$

which only depends on the fraction of treated neighbors: $\frac{\sum_{m \neq k} E_{km} w_m}{\sum_{m \neq k} E_{km}}$. Thus this assumption is properly satisfied.

**Rationality of Assumption 4:** From the proof of rationality of Assumption 3 (Anonymous interference),

$$f_k(w, x) = \frac{\lambda}{N} \sum_{i} \sum_{\gamma} \phi_{\gamma,i} \frac{\alpha_{\gamma,i}(\theta_k, w) v_{\gamma,i}(\theta_k, w)}{\epsilon_{\gamma,i} + \sum_{\theta'} G_N(\alpha_{\gamma,i}(\theta', 1) v_{\gamma,i}(\theta', 1) x + \alpha_{\gamma,i}(\theta', 0) v_{\gamma,i}(\theta', 0)(1 - x))}.$$

Since $\epsilon_{\gamma,i}$ is non-negative, $\alpha_{\gamma,i}(\theta', 1)$, $v_{\gamma,i}(\theta', 1)$, $G_N$, $\alpha_{\gamma,i}(\theta', 0)$, $v_{\gamma,i}(\theta', 0)$ are positive for all $\gamma$ and $i$, $|f(w, x)|$, $|f'(w, x)|$, $|f''(w, x)|$, $|f'''(w, x)|$ are bounded uniformly in $w \in \{0, 1\}$ and $x \in [0, 1]$, thus this assumption is properly satisfied.

## A.3 COMPARATIVE EXPERIMENTS WITH EXISTING METHODS

Figure 5 shows the comparison of our estimator with the customer-side estimator, listing-side estimator, and TSR-Naïve estimator proposed in Johari et al. (2022) in the homogeneous setting, where $\lambda = \tau = 1$. Customers have utility $v = 0.315$ for control listings and $\widetilde{v} = 0.394$ for treatment listings.

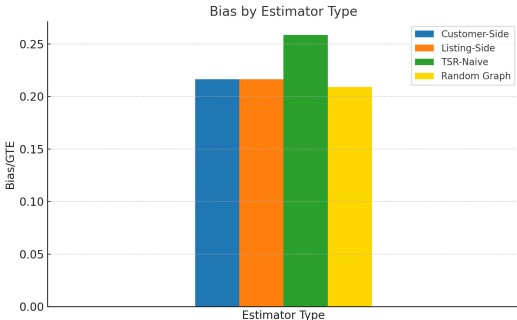

Figure 5: Comparison of Estimators

