# OpenReview forum: "Random Graph Asymptotics for Treatment Effect Estimation in Two-Sided Markets"
_ICLR.cc/2025/Conference — Submitted to ICLR 2025_

### Official Review · Reviewer_nxGk · 2024-10-28

**Soundness:** 3
**Presentation:** 1
**Contribution:** 1
**Rating:** 3
**Confidence:** 3

**Summary:**

This paper investigates a critical issue in two-sided markets that does not satisfy the Stable Unit Treatment Value Assumption (SUTVA). It introduces a novel framework that utilizes random graph asymptotics to model and estimate treatment effects under network interference in these markets.

**Strengths:**

- This paper addresses a significant issue with practical applications and academic value.

- The introduced method is theoretically sound and supported by empirical validation, providing valuable guidance for future research on random graph asymptotics in two-sided markets.

**Weaknesses:**

- My primary concern is the paper's limited contribution to the causal reasoning community. It appears to apply the methods and theories from [1] to the specific context of two-sided markets without offering unique technical advancements.

- Additionally, while the paper acknowledges its foundation in [1], it contains significant similarities in text, formulas, and figures, raising concerns about potential plagiarism.

- The main distinction from [1] lies in its application to two-sided markets. However, the lack of validation with real-world data, relying solely on simulations, further diminishes its contribution.

- If the authors can articulate the key differences between their work and that of [1], along with the significant challenges these differences present, and demonstrate how their method addresses these challenges with real-world validation in two-sided markets, I would reconsider my score.

---

[1] Li S, Wager S. Random graph asymptotics for treatment effect estimation under network interference[J]. The Annals of Statistics, 2022, 50(4): 2334-2358.

**Questions:**

as above.

---

> ### Author Response · Authors · 2024-11-25
>
> Dear Reviewer nxGk,
>
> Thank you for your thoughtful review of our work. We appreciate your recognition of the practical and academic significance of addressing treatment effect estimation under interference in two-sided markets, as well as your acknowledgment of the theoretical soundness and empirical validation of our framework. Below, we address your concerns in detail.
>
> **Q1. Contribution to the Causal Reasoning Community**
>
> While our framework builds on the foundational work of Li and Wager’s, our contributions extend their methods to the unique and important setting of two-sided markets. We model the interference on both sides of the market (listings and customers) under two-sided randomization, introducing a more complex randomization and interference structure than the single-sided setting considered in Li and Wager’s. Specifically, extending Li and Wager’s results to our context required addressing two major challenges:
>
> - Incorporating a random graph model in the two-sided randomization setting. Li and Wager’s results are limited to a single set of treated units. Initially, we considered including both the listing side and the customer side in the random graph model since both sides receive treatments and controls. However, upon carefully analyzing the mechanics, we concluded that the interference between customers is more appropriately modeled within the potential outcome function (Eq. 2) as a function of both graph topology and equilibrium dynamics, and the interference between listings is best captured through the random graph model.
>
> - Determining the specific form of the graphon. Unlike the static setting in Li and Wager’s work, our framework necessitates a dynamic interference model. To address this, we focused on the steady-state environment. The graphon represents the probability of two listings being connected, accounting for the interference between them. Since interference only occurs among listings available at the same time, we use the steady-state mass of listings to determine the graphon value in Eq. 1. Our work incorporates equilibrium dynamics, which are critical for two-sided markets but absent in Li and Wager’s. This required adapting their methodology to account for steady-state circumstance, as well as deriving new variance adjustments for treatment effect estimation.
>
> We revised the “Our Contributions” section to better highlight these distinctions, technical challenges and the methodological decisions they required in extending Li and Wager’s framework to the two-sided market context.
>
> **Q2. Concerns About Similarities**
>
> We appreciate your concern about similarities with Li and Wager’s work. Our paper explicitly acknowledges Li and Wager’s as a key reference. We cite their paper properly multiple times, made our own contribution clear and properly cites their contributions throughout. While our work builds on their framework, we have introduced significant methodological advancements, as detailed in the previous section. Some overlap in text arises from our adaptation of their assumptions and random graph model, which is common in academic writing when building on existing frameworks. Similarities in figures stem from using the same criteria to assess the estimator, but the code for generating our figures is original and available in the supplementary materials. While some formulas share structural similarities with those in Li and Wager’s work, they have been tailored to suit the two-sided market context. The revision of our manuscript explicitly highlights our methodological novelties and ensures that our contributions are distinct, clearly stated, and appropriately credited.
>
> Thank you again for your constructive feedback. Your comments have been invaluable in identifying areas for improvement. Please let us know if there are any additional points you would like us to address.

---

> > ### Comment · Reviewer_nxGk · 2024-11-28
> >
> > Thank you for your response, which addresses several of my concerns. However, the current version of the paper does not fully articulate the points you mentioned. The revisions only make brief changes in the introduction, without providing a detailed comparison in the main text between the single-sided and two-sided settings or between static and dynamic interference scenarios. Additionally, the paper does not sufficiently highlight the theoretical and practical advantages of your method in the two-sided and dynamic interference settings compared to the approach by Li and Wager. If the revised paper includes a thorough theoretical and experimental comparison with their work, I will consider raising my score.

---

> > > ### Author Response · Authors · 2024-11-28
> > >
> > > Dear Reviewer nxGk,
> > >
> > > Thank you very much for your thoughtful feedback. Although the time available to incorporate revisions before the submission deadline was limited, we prioritized addressing your suggestions and tried our best to revise the paper accordingly. We updated the "Random Graph Model" section to better clarify our contributions on two-sided randomization (lines 190–196, 251–260) and dynamic interference (lines 215–246).
> > >
> > > We also understand your concern regarding comparisons with Li and Wager’s approach. As their method is relatively general and not tailored to a specific setting, while ours focuses on two-sided markets, a direct comparison is challenging. However, [1] also focuses on two-sided platforms. To address this, we compared the performance of our estimator with the customer-side estimator, listing-side estimator, and TSR-Naïve estimator proposed in [1], which can be found in Appendix A.3.
> > >
> > > We deeply appreciate your constructive suggestions and hope that these updates effectively address your concerns.
> > >
> > > ---
> > >
> > > [1] Ramesh Johari, Hannah Li, Inessa Liskovich, and Gabriel Y. Weintraub. Experimental Design in Two-Sided Platforms: An Analysis of Bias. Management Science, 2022.

---

### Official Review · Reviewer_d4Pv · 2024-11-02

**Soundness:** 3
**Presentation:** 3
**Contribution:** 2
**Rating:** 3
**Confidence:** 5

**Summary:**

The manuscript has a rigorous mathematical framework for interference, and clear asymptotic properties. I liked the novel use of graphon models. Authors offer estimations for both direct and indirect effects with cariance adjustments for interference. The proposed methods is computationally tractable.

Aside from the rigor and nice theoretical set up, and for the scope of ICLR, I found the method challenging to use in real applications. The experimental setup require two-sided randomization and clean treatment assignment with complete tracking of interactions. This seems like a tall call, as in reality, we have multiple concurrent changes with partial observability. Also authors did not focus on a real wold domain, but the assumptions of poisson arrivals, multinomial logit choice, and xponential occupancy times often violated and replaced by seasonal/time-varying patterns, variable occupancy durations, and heterogeneous listing attributes.

For such setting, I suggest discussing adaptations to observational setting with possible extensions:

- Incorporate propensity score methods

- Add covariate adjustment

- Model selection bias

Implementation:

- Match similar time periods

- Control for external factors

- Use proxy variables for unobserved confounders

Other than observational case, I also suggest discussing A/B Testing Framework, with some modification in design:

1. Simplified Treatment:

   - Single-sided treatment first

   - Gradual rollout

   - Clear control groups

2. Measurement:

   - Track spillover effects

   - Monitor market equilibrium

3. Analysis:

   - Compare with benchmark A/B estimates, and possibly with available real-world data

   - I also, like to see validation on the interference patterns (maybe in single market segment with limited time period?, what can be good diagnostic tests for interference?)

**Strengths:**

see summary

**Weaknesses:**

see summary

**Questions:**

see summary

---

> ### Author Response · Authors · 2024-11-25
>
> Dear Reviewer d4Pv,
>
> Thank you for your thorough review and insightful feedback on our submission. We greatly appreciate your recognition of the rigor of our mathematical framework, the novel use of graphon models, and the computational tractability of the proposed methods. Your comments have provided valuable insights and highlighted potential extensions to our work.
>
> **Q1. Adaptation to Observational Settings**
>
> Your suggestion to extend our framework to observational settings is highly valuable and aligns with potential directions for future research. While our current method is tailored for randomized experiments, adapting it for observational data represents a natural extension of our work. Propensity score methods and covariate adjustments could be integrated to account for confounding; potential methods such as the use of proxy variables could address selection bias and unobserved confounders; matching similar time periods and controlling for external factors could improve the validity of observational estimates. We indicated these possibilities in the conclusion section as future research directions in the revised manuscript.
>
> **Q2. A/B Testing Framework**
>
> We appreciate your suggestions regarding the design of an A/B testing framework. While this lies beyond the scope of our current study, your proposed ideas, such as simplified treatments, gradual rollouts, and tracking spillover effects, provide valuable guidance for translating our framework into practical applications. We also highlighted these points as future research directions in the revised manuscript.
>
> **Q3. Practical Applicability of the Method**
>
> We acknowledge that real-world markets exhibit complexities such as seasonality, variable durations, and heterogeneous attributes. However, the assumptions in our framework, including Poisson arrivals, multinomial logit choice, and exponential occupancy times, serve as well-established and tractable approximations that allow us to focus on the core dynamics of interference and treatment effects. These simplifications are widely used in the literature to provide a theoretical foundation, which can be iteratively refined in future work. Extensions to incorporate time-varying patterns or more flexible models are promising future directions that build on our current framework without altering its core contributions.
>
> Once again, we thank you for your constructive feedback. While the suggestions you provided represent valuable directions for future research, they require substantial modifications that are beyond the current scope of this submission. We look forward to exploring them in future work. Please feel free to let us know if additional clarifications are needed.

---

> > ### Comment · Reviewer_d4Pv · 2024-12-02
> >
> > I thank the authors for their responses. However, all my critics is postponed to future work, that being said, I stand by my previous rating.

---

> > > ### Author Response · Authors · 2024-12-03
> > >
> > > Dear Reviewer d4Pv,
> > >
> > > Thank you for your feedback. We appreciate your suggestion to address them in future work.

---

### Official Review · Reviewer_zKRQ · 2024-11-02

**Soundness:** 3
**Presentation:** 3
**Contribution:** 3
**Rating:** 6
**Confidence:** 4

**Summary:**

The paper studies treatment effect estimation in two-sided markets, specifically the listing markets. The market dynamics is the following. A customer arrives, forms consideration set using currently available listings ($a_\gamma(\theta)$), and choose one listing or not choosing at all based on the utility $v_\gamma(\theta)$. By assuming a Poisson process on costumers, the listing status is a continuous time markov chain. The equilibrium is used to define the potential outcome of listings. Also an interference structure based on graphs is imposed on the potential outcome in addition to the equilibrium structure. Asymptotically normal estimators for direct and indirect effects are proposed.

**Strengths:**

1. The listing dynamics is practical for many online platforms. The paper studies treatment effect estimation under the equilibrium effect of the platform, which is relevant for practical purposes.

**Weaknesses:**

1. It may be helpful to outline technical difficulties when extending Li and Wager's results to highlight the paper's theoretical contributions.

2. Unless I'm missing something, constant graphon function in Eq (1) is restrictive compared to the previous Li and Wager's results. If the paper is assuming constant graphon,

**Questions:**

Line 380: "regularization assumptions" or "regularity assumption"?

Line 066: inconsistent citation format.

Line 167: may be better to repeat definition of Y_i here; it is definition is hidden in a long paragraph.

Line 225: i and j were used to index listings and buyers previously.

In Eq(2), is the form of potential outcome outcome function the assumption? Because after eq (2) it says "derivation", which is confusing. Also the meaning of the potential outcome function should be explained too. I believe it represents the steady-state utility the listings are generating. Also Eq (2) should be defined right after Section 2 to make it clear that the equilibrium described in Sec 2 is used to define potential outcomes.

main questions

1. In Eq (1), the RHS is the same for all theta_i, theta_j. Is it a typo? The original Li and Wager paper allows for general G if i'm not missing something. A constant graphon function restricts the class of treatment interference being modeled.

2. Are there technical difficulties when extending results of Li and Wager to the listing dynamics?

3. What are the data we observe to estimate treatment effects? Just the outcomes $Y$? If we additionally observe consumer types and values $v$ and consideration probability $\alpha$, how would a model-based approach behave compared to the proposed method? It could be that I'm missing something, to me both estimators for direct and indirect effects are directly borrowed from Li and Wager, and do not use the rich dynamics assumed.

I would be happy to update score if the authors could sufficiently address these questions.

---

> ### Author Response · Authors · 2024-11-25
>
> Dear Reviewer zKRQ,
>
> Thank you very much for your thoughtful and detailed review of our work. We deeply appreciate the time and effort you took to provide constructive feedback, which has been instrumental in helping us refine and clarify our contributions. We are pleased that you found our modeling of listing dynamics and equilibrium effects practical and relevant for real-world applications. Your acknowledgment of the significance of studying treatment effects in two-sided markets further motivates us to enhance the quality of our work.
>
> **Q1. Technical Challenges in Extending Li and Wager’s Results**
>
> Extending Li and Wager’s results to our setting required addressing two major challenges:
>
> - Incorporating the random graph model in the two-sided randomization setting. Li and Wager’s results are limited to a single set of treated units. Initially, we considered including both the listing side and the customer side in the random graph model since both sides receive treatments and controls. However, upon carefully analyzing the mechanics, we concluded that the interference between customers is more appropriately modeled within the potential outcome function (Eq. 2) as a function of both graph topology and equilibrium dynamics, and the interference between listings is best captured through the random graph model.
>
> - Determining the specific form of the graphon. Unlike the static setting in Li and Wager’s work, our framework necessitates a dynamic interference model. To address this, we focused on the steady-state environment. The graphon represents the probability of two listings being connected, accounting for the interference between them. Since interference only occurs among listings available at the same time, we use the steady-state mass of listings to determine the graphon value in Eq. 1.
>
> We revised the “Our Contributions” section to better highlight these challenges and the methodological decisions they required.
>
> **Q2. Constant Graphon Function in Eq. 1**
>
> You are correct that using a constant graphon simplifies the interference structure, and a more nuanced model with graphon values determined by listing types is a promising direction for future work. However, as we discussed in the last question, interference effects are most influenced by whether a listing is available rather than its specific type. Thus, our choice of a constant graphon suffices to derive interpretable and tractable estimators under equilibrium dynamics.
>
> **Q3. Line-by-Line Issues**
>
> We have revised the manuscript to address the following formatting and clarity issues:
>
> Line 068 (former line 66): Corrected the citation format.
>
> Line 179 (former line 167): Repeated the definition of Y_i for clarity.
>
> Line 227 (former line 225): Used distinct indices for the treatment conditions of listings and buyers.
>
> Line 379 (former line 380): Replaced “regularization assumptions” with “regularity assumption.”
>
> These changes improve the consistency and readability of the manuscript, and we sincerely thank you for carefully reading it and pointing them out.
>
> **Q4. Explanation of Eq. 2**
>
> We agree that Eq. 2 could be introduced more clearly. The form of the potential outcome function is not an assumption but is derived, as detailed in Appendix A.1 of our revised manuscript. To improve clarity, we explicitly define it as the steady-state booking rate. However, while the potential outcome function is derived from the equilibrium described in Section 2, its specific form as a function of the fraction of treated neighbors relies on Assumption 3, and its simplification also depends on Assumption 2. For this reason, we can’t place it earlier in the manuscript.

---

> > ### Author Response · Authors · 2024-11-25
> >
> > **Q5. Clarification of Data and Rich Dynamics**
> >
> > - While our estimators are inspired by Li and Wager’s methodology, we extend their framework by incorporating equilibrium-based dynamics and graph-based interference. In real-world two-sided markets, outcomes Y are often much more accessible than other parameters. Our method suffices to work effectively under these conditions and provides a robust estimation of treatment effects.
> >
> > - The rich dynamics assumed in our model are crucial for deriving the potential outcome function in Eq. 2. The derived form of the potential outcome function enables us to validate the rationality of Assumption 3 (anonymous assumption) in our two-sided market setting, which is a key component of Li and Wager’s results. This ensures that their methodology is appropriately adapted and applied to our context.
> >
> > - If richer data such as consumer types or consideration probabilities becomes available, we hypothesize that model-based approaches could potentially outperform the proposed method. We plan to explore this direction in future work.
> >
> > Once again, we appreciate your valuable feedback and encouragement. Your comments have significantly improved the rigor and clarity of our paper. We hope that the revised version will address your concerns comprehensively, and we are happy to provide further clarifications if needed.

---

> > > ### Comment · Reviewer_zKRQ · 2024-11-26
> > > **Thank you for the response**
> > >
> > > I thank the authors for the clarification. Since the paper proposes a new problem setup, it would be important to investigate the simple baseline (model-based methods), and then compare either theoretically or empirically to show what is the benefit of the new method. I keep my original score.

---

> > > > ### Author Response · Authors · 2024-11-28
> > > >
> > > > Dear Reviewer zKRQ,
> > > >
> > > > Thank you very much for your thoughtful comments and feedback. We agree that comparing our proposed method with a simple baseline can demonstrate the benefits of our approach. Due to space and time constraints, we were unable to include such a comparison in the current submission. Nevertheless, we believe this is an excellent direction for future work.
> > > >
> > > > Thank you again for your insightful suggestion and for taking the time to review our work.

---

### Official Review · Reviewer_8dZA · 2024-11-11

**Soundness:** 3
**Presentation:** 2
**Contribution:** 3
**Rating:** 6
**Confidence:** 2

**Summary:**

This paper tackles the challenging problem of estimating treatment effects in two-sided markets, a context where interference between market participants poses significant obstacles to causal inference. In which would violate the Stable Unit Treatment Value Assumption. The paper proposes a method that leverages random graph asymptotics to address these challenges.

**Strengths:**

1. The accurate estimation of treatment effects is essential but challenging due to network interference.
2. The proposed method offers a novel solution by utilizing random graph asymptotics.
3. The numerical simulations demonstrate the practical applicability and effectiveness of the method.

**Weaknesses:**

1. The paper could benefit from a more detailed comparison with existing methods to highlight the advantages and limitations of the proposed method.
2. It would be better to add comparative experiments with existing methods in the experimental section.

**Questions:**

See above.

---

> ### Author Response · Authors · 2024-11-25
>
> Dear Reviewer 8dZA,
>
> Thank you very much for your constructive review of our work. We greatly appreciate your recognition of the challenges posed by network interference in two-sided markets and your acknowledgment of the novelty of our proposed method utilizing random graph asymptotics. Additionally, we are glad that the practical applicability and effectiveness of our method, as demonstrated through numerical simulations, resonated with you. Your feedback has provided valuable insights that will help us improve the clarity and comprehensiveness of our paper.
>
> **Q1. Comparison with Existing Methods**
>
> - We agree that a more detailed comparison with existing methods would better highlight the advantages of our approach. In the “Related Works” section of our manuscript (lines 59–70), we indicate that in the existing methods, their estimators are particularly sensitive to the relative volumes of supply and demand, leading to biases due to the interconnected nature of participants. Furthermore, these methods lack desirable asymptotic properties and only provide point estimates without confidence intervals.
>
> - In contrast, our approach addresses these limitations by not only offering confidence intervals but also demonstrating robust asymptotic properties. Our estimator for the direct effect is asymptotically normal around the true direct effect in the presence of interference, and we quantify the excess variance introduced by interference effects. The PC-balancing estimator for the indirect effect is both consistent and asymptotically normal.
>
> - In our revised manuscript, we updated the “Our Contributions” section to provide a clearer understanding of our method's contributions and its position relative to existing work.
>
> **Q2. Comparative Experiments with Existing Methods**
>
> - We also agree that adding comparative experiments would strengthen our paper. In the Appendix A.3 of the revised manuscript, we include comparisons of our estimator with the customer-side estimator, listing-side estimator, and TSR-Naïve estimator proposed in [1].
>
> Once again, we sincerely thank you for your thoughtful comments and encouragement. Please do not hesitate to reach out if further clarifications are needed.
>
> ---
>
> [1] Ramesh Johari, Hannah Li, Inessa Liskovich, and Gabriel Y. Weintraub. Experimental design in two-sided platforms: An analysis of bias. Management Science, 2022.

---

### Note · Authors · 2025-02-05

I have read and agree with the venue's withdrawal policy on behalf of myself and my co-authors.

---

### Meta-Review · Area_Chair_7mcV · 2024-12-20

**Metareview:**

This paper looks at some causal inference in markets where users arrive sequentially.

Several issues were raised by the different reviewers, and the authors acknowledge that they will taken care of in future work. I guess this means in the next submission, where a revised version of this paper would be much more interesting.

Unfortunately, the paper is not ready right now for publications.

**Additional Comments On Reviewer Discussion:**

The authors even agree with the reviewers that this paper needs polishing and will be ready at another round of submissions. No real discussion was needed

---

### Decision · Program_Chairs · 2025-01-22

Reject